# From a Clustering of Adverse Symptoms after Colorectal Cancer Therapy to Chronic Fatigue and Low Ability to Work: A Cohort Study Analysis with 3 Months of Follow-Up

**DOI:** 10.3390/cancers16010202

**Published:** 2024-01-01

**Authors:** Tomislav Vlaski, Marija Slavic, Reiner Caspari, Bettine Bilsing, Harald Fischer, Hermann Brenner, Ben Schöttker

**Affiliations:** 1Division of Clinical Epidemiology and Aging Research, German Cancer Research Center (DKFZ), 69120 Heidelberg, Germany; tomislav.vlaski@dkfz-heidelberg.de (T.V.); marija.slavic@dkfz-heidelberg.de (M.S.); h.brenner@dkfz-heidelberg.de (H.B.); 2Medical Faculty Heidelberg, Heidelberg University, 69117 Heidelberg, Germany; 3Clinic Niederrhein, 52474 Bad Neuenahr-Ahrweiler, Germany; reiner.caspari@klinik-niederrhein.de; 4Clinic Bad Salzelmen, 39218 Schönebeck, Germany; bettine.bilsing@wz-kliniken.de; 5Clinic Rosenberg, 33014 Bad Driburg, Germany; harald.fischer@drv-westfalen.de; 6National Center for Tumor Diseases (NCT), German Cancer Consortium (DKTK), German Cancer Research Center (DKFZ), 69120 Heidelberg, Germany

**Keywords:** colorectal cancer, rehabilitation, fatigue, ability to work, symptom clusters

## Abstract

**Simple Summary:**

After getting treated for colorectal cancer, many people feel fatigue, stress, and pain and face other challenges that can make daily activities and work tough. Our research explored how these issues group together and if they are connected to fatigue and difficulties in work. We gathered data from the MIRANDA study, where patients shared their experiences after colorectal cancer treatment before the start of the rehabilitation and three months after. Using data from their questionnaire responses, we identified six main problem groups: fatigue, digestive troubles, pain, feelings of stress or anxiety, urinary issues, and side effects from chemotherapy. Our findings reveal that experiencing even a single symptom from these categories can be a precursor to feeling more fatigued or having a decreased ability to work. Our findings shed light on colorectal cancer survivors’ various challenges and how they might impact their daily lives and work.

**Abstract:**

In colorectal cancer (CRC) patients, apart from fatigue, psychological and physical symptoms often converge, affecting their quality of life and ability to work. Our objective was to ascertain symptom clusters within a year following CRC treatment and their longitudinal association with persistent fatigue and reduced work ability at the 3-month follow-up. We used data from MIRANDA, a multicenter cohort study enrolling adult CRC patients who are starting a 3-week in-patient rehabilitation within a year post-curative CRC treatment. Participants completed questionnaires evaluating symptoms at the start of rehabilitation (baseline) and after three months. We performed an exploratory factor analysis to analyze the clustering of symptoms at baseline. Longitudinal analysis was performed using a multivariable linear regression model with dichotomized symptoms at baseline as independent variables, and the change in fatigue and ability to work from baseline to 3-month-follow-up as separate outcomes, adjusted for covariates. We identified six symptom clusters: fatigue, gastrointestinal symptoms, pain, psychosocial symptoms, urinary symptoms, and chemotherapy side effects. At least one symptom from each factor was associated with higher fatigue or reduced ability to work at the 3-month follow-up. This study highlights the interplay of multiple symptoms in influencing fatigue and work ability among CRC patients post-rehabilitation.

## 1. Introduction

Cancer-related fatigue is a pervasive and debilitating symptom experienced by many colorectal cancer (CRC) patients, which can persist for months or even years after primary cancer therapy [1]. Besides fatigue, CRC survivors also commonly report other symptoms, such as depression, anxiety, pain, sleep disturbance, digestive tract symptoms, and sexual dysfunction [2,3,4]. These symptoms arise from the physical and psychological sequelae of CRC, influenced by its various treatments, including chemotherapy, radiotherapy, and surgical interventions. These therapeutic approaches, while targeting oncological control, concurrently contribute to the spectrum of symptomatology in CRC patients, ranging from fatigue to psychosomatic distress. 

Chemotherapy is one of the vital treatment components in stage II or III CRC patients. However, the inherent nature of chemotherapy results in not only the destruction of cancer cells but also damage to healthy cells, leading to various adverse effects [5]. CRC patients undergoing chemotherapy experience adverse effects that impact their physical health, overall quality of life [4,5,6], and emotional state [6]. A study from Australia examining the adverse effects of chemotherapy in routine care found that CRC patients had the highest prevalence of fatigue (88%), with 89% of CRC patients having at least one adverse effect [7]. 

Radiotherapy has become an integral part of managing rectal cancer, contributing to a significant reduction in the rate of local recurrences when paired with total mesorectal excision surgery in stage II and III CRC patients [8]. Notably, neoadjuvant radiotherapy, administered before surgery, has been associated with lower local recurrence rates and fewer toxic effects than adjuvant therapy given after the operation [9]. While radiotherapy is an effective treatment strategy for rectal cancer, it is not without potential side effects due to the impact on surrounding organs. Locally, patients may experience diarrhea, frequent urination, gas and bloating, cramping, skin irritation, and constipation [10]. In addition, systemic side effects like fatigue and loss of appetite might significantly impact the patient’s well-being [11].

Previous studies have reported that multiple symptoms cluster together in CRC survivors, suggesting that these symptoms may be interconnected and share underlying mechanisms [12]. For example, one study identified four distinctive symptom clusters (psychological, digestive and urinary, low energy, and pain) in CRC patients during the first year after surgery [13]. Functional status and quality of life are often significantly impaired among CRC survivors who experience these adverse symptoms simultaneously [14,15]. Therefore, modifying the traditional single-symptom approach and addressing symptom clusters when treating CRC patients might be beneficial.

The return to work of cancer patients is an essential milestone in rehabilitation after primary treatment. Patients who return to work report regaining a sense of normalcy and purpose, which is closely linked to the ability to provide for oneself and economic independence [16,17]. However, only 57% of cancer patients have been reported to return to work after treatment [18]. The symptoms of primary disease and adverse effects stemming from associated treatments often form significant barriers to resuming professional activities. Fatigue and depression can cause decreased productivity, while physical symptoms lead to lower employment rates [19]. Therefore, the challenges do not solely arise from the cancer diagnosis but are also intricately related to the treatment side effects, further complicating the patient’s ability to reintegrate into the pre-diagnosis work environment.

As fatigue commonly co-occurs with various adverse effects of CRC treatment, it is crucial to investigate the association between adverse symptoms and persistent cancer-related fatigue and low ability to work with a longitudinal study design. With a better understanding of the temporal relationship, it can be better estimated whether such adverse symptoms are causally related to fatigue and ability to work or are simply bystanders of the associations of CRC treatments and fatigue. 

The first aim of this study was to determine whether there are clusters of symptoms in CRC patients undergoing in-patient rehabilitation. The investigated symptoms include psychological, physical, and fatigue-related symptoms and issues impacting daily life and social interactions. The second aim was to explore which of these symptoms are also associated with changes in fatigue and the ability to work in the first three months after rehabilitation.

## 2. Materials and Methods

### 2.1. Study Design and Participants

The MIRANDA study is a multicentric, prospective cohort study with the aim to investigate risk, preventive, and prognostic factors of fatigue, QoL, ability to work, and other health-related outcomes among CRC patients during and after in-patient rehabilitation [20]. Participants are recruited in 6 rehabilitation clinics located in different regions in Germany to ensure a representative sample of the German rehabilitation setting. Besides an age of 18 years and older, the inclusion criteria for the MIRANDA study were CRC diagnosis, CRC therapy (surgery, radiation, and/or chemotherapy) in the last 12 months and sufficient knowledge of the German language. No written informed consent was the only exclusion criterion. The recruitment phase began in September 2020 and will continue until the end of 2025 with the objective of enrolling 1000 study participants.

During the initial week of their in-patient rehabilitation (baseline), participants complete a baseline questionnaire and donate blood and stool samples. They are then required to fill out a self-administered follow-up questionnaire every 3 months during the first year and subsequently at the 3rd, 5th, 7th, and 10th year.

In the context of the MIRANDA study, participants who have 25-hydroxyvitamin D levels below 60 nmol/L and are deemed suitable for vitamin D supplementation are given the opportunity to participate in the “Personalized Vitamin D supplementation for reducing or preventing fatigue and enhancing the quality of life of patients with colorectal tumor–randomized intervention trial” (VICTORIA) [21]. Approximately 50% of the participants involved in the MIRANDA study are also taking part in the VICTORIA trial, in which they are randomly selected to undergo a 12-week regimen of either vitamin D_3_ supplementation or a placebo.

### 2.2. Study Measurements

#### 2.2.1. Demographic, Lifestyle, and Clinical Data

Sociodemographic, lifestyle, and clinical data were collected from the participant’s baseline questionnaires. Sociodemographic data include sex and age, and lifestyle data include physical activity levels, smoking status, and the body mass index (BMI). Clinical data encompass the stage of disease, mode of treatment (surgery, chemotherapy, or radiotherapy), months since CRC surgery, and the number of comorbidities. All of the sociodemographic, lifestyle, and clinical data were self-reported, apart from weight and height, which were measured at baseline and cancer stage, which was physician-reported. 

#### 2.2.2. Symptom Scales

All the symptom scales utilized in this study are validated instruments, ensuring accurate and consistent measurement of patient experiences. Fatigue and the ability to work were estimated using the Functional Assessment of Chronic Illness Therapy–Fatigue–Fatigue Scale (FACIT-F-FS) score and the FACIT-F-Functional Well-Being–Ability to Work (FACIT-F-FWB-AW) item, respectively [22]. FACIT-F-FS is scaled from 0 to 52, with a higher score denoting less fatigue. Cancer-related fatigue was defined as having a score of less than 34 on the FACIT-F-FS, previously established as clinically relevant fatigue [23]. FACIT-F-FWB-AW item is scored from 1 to 5, with a higher score denoting better ability to work. A low-to-moderate ability to work is defined by the response categories “Not at all” and “A little bit” to the FACIT-F-FWB-AW item (“I am able to work, including work at home”) (score ≤ 2 points) [20].

Pain, fatigue, nausea/vomiting, dyspnea, insomnia, appetite loss, constipation, financial difficulties, and diarrhea were ascertained using the European Organization for the Research and Treatment of Cancer–Quality of Life Questionnaire–Core 30 (EORTC-QLQ-C30) symptoms scale [24]. Symptoms are scaled from 0 to 100, with a higher score indicating greater symptom presence. The following cut-off values were used to define a clinically significant symptom presence: fatigue > 39 points, pain > 25 points, nausea/vomiting > 8 points, insomnia > 50 points, dyspnea > 17 points, appetite loss > 50 points, constipation > 50 points, diarrhea > 17 points, and financial difficulties > 17 points [24].

Urinary frequency, blood and mucus in stool, stool frequency, urinary incontinence, dysuria, abdominal pain, buttock pain, bloating, dry mouth, hair loss, taste, flatulence, fecal incontinence, sore skin, embarrassment, and impotence/dyspareunia were assessed using the European Organization for Research and Treatment of Cancer Quality of Life Questionnaire-Colorectal Cancer Module (EORTC QLQ-CR29) [25]. Each symptom was rated in terms of how often it appeared during the previous week, ranging from 1 (not at all) to 4 (very much), and transformed into a linear scale from 0 to 100. For the current study, the EORTC QLQ-CR29 symptoms were dichotomized into *none* or *a little bit* (answer category 1 or 2) versus *moderate* and *a lot* (answer category 3 or 4). For those symptoms where the number of participants in the second category was below 20%, the answer category *a little bit* was merged with the *moderate* and *a lot* categories. Combining these categories for less prevalent symptoms acknowledges that even little symptom expression can have a substantial impact on CRC patients’ well-being and quality of life.

Physical fatigue, emotional fatigue, cognitive fatigue, interference with daily life, and social sequelae were assessed using the European Organization for Research and Treatment of Cancer Quality of Life Questionnaire–Fatigue Module 12 (EORTC QLQ-FA12) questionnaire [26]. In order to determine the cut-off values for the EORTC QLQ-FA12 subscales, the General Fatigue Index was calculated using physical, emotional, and cognitive fatigue scales, ranging from 0 to 30 points, with ≥12 points denoting fatigue levels that require clinical assistance [27]. Given that the prevalence of clinically relevant fatigue was close to 50% (57% using the FACIT-F-FS scale and 53% using the Global Fatigue Index), we used the median (50 points) as a cut-off value for the EORTC-QLQ-FA12 subscales. 

Psychosocial distress was estimated using the Questionnaire on Stress in Cancer Patients (QSC-R10). The QSC-R10 uses a five-point Likert scale ranging from 0 (not at all) to 4 (very much) to assess the frequency and intensity of distress experienced by cancer patients. The total score ranges from 0 to 40, with higher scores indicating higher levels of distress. A cut-off score of >14 points has been established to determine the need for psychosocial support [28]. 

### 2.3. Statistical Analysis

Participants who completed their baseline questionnaires between 09/2020 and 10/2023 were included in the analyses. For the longitudinal analyses, a subset of participants who had completed 3-month follow-up was selected.

Descriptive statistical measures were used to characterize the baseline characteristics of the participants and their fatigue and ability to work at baseline and 3-month follow-up. Pearson’s chi-squared test was used to compare the participants who completed the 3-month follow-up, those who did not respond, and those who were recruited late.

Symptom clusters were identified through exploratory factor analysis. All symptom scales from the validated FACIT-F-FS, FACIT-F-FWB-AW, EORCT-QLQ-C30, EORTC QLQ-CR29, EORTC-QLQ-FA12, and QSC-R10 were used as defined in Section 2.2.2 with only few exceptions. Impotence/dyspareunia and stoma care were not included due to a high percentage of missing values. Fatigue from the EORCT-QLQ-C30 and physical fatigue from the EORCT-QLQ-FA12 were excluded due to high correlation with the FACIT-F-FS fatigue item (r = −0.86 and r = −0.86, respectively). The final number of symptoms included in the exploratory factor analysis was 32. The Kaiser–Meyer–Olkin measure was employed to assess the dataset’s suitability for factor analysis. Oblique rotation with the promax option was applied to simplify the factor structure. The final number of factors was determined using the Minimum Average Partial and Parallel Analysis methods.

The longitudinal analysis was performed with a multiple linear regression model. Assumptions of multiple linear regression (linearity, independence, multicollinearity, homoscedasticity) were tested prior to the analysis and were met for all analyses [29]. Symptoms that exhibited factor loadings above 0.3 in the exploratory factor analysis were dichotomized and selected as independent variables. Longitudinal changes in fatigue (defined as FACIT-F-FS 3-month follow-up scale subtracted from its baseline value) and ability to work (defined as FACIT-F-FWB-AW score at the 3-month follow-up subtracted from its baseline value) were used as dependent variables. The regression models for these two outcomes were adjusted for baseline FACIT-F-FS and FACIT-F-FWB-AW scales, respectively. Furthermore, all models were adjusted for age, sex, CRC stage, months since CRC surgery, chemotherapy and/or radiotherapy in the last 12 months, BMI, baseline smoking status, physical activity before surgery, and number of comorbidities. In subgroup analysis, analyses were stratified by sex, age groups (<65 years/≥65 years), and additional participation in the VICTORIA trial to corroborate if this potentially impacted the results.

To impute missing values of covariates at baseline, multiple imputation by the Markov Chain Monte Carlo method was applied. Twenty-five datasets were imputed separately with the SAS procedure PROC MI. To our knowledge, missing values of other covariates are following the “missing at random” pattern. All longitudinal analyses were performed in the 25 imputed datasets, and results were combined by the SAS procedure PROC MIANALYZE. The covariate matrix for the exploratory factor analysis was imputed using the “mifa” package in R [30].

All analyses were performed using SAS software version 9.4, except for the exploratory factor analysis performed in R, version 4.2.3 (package “psych”). A two-sided significance level of *p* < 0.05 was used for all tests. 

## 3. Results

### 3.1. Baseline Characteristics

In total, 394 participants filled out the baseline questionnaire, of whom 279 returned a 3-month follow-up questionnaire. Among the non-responders, 64 were recruited in the last 3 months and did not have a chance to respond yet. Not counting these study participants results in a response rate at the 3-month follow-up of 84.5%. When comparing the participants who completed the 3-month follow-up questionnaire to those who were later recruited and those who did not respond by age, sex, and cancer stage, no significant differences were found (Appendix A).

The mean age of the study participants was 63 years, and 57.9% were male (Table 1). While the CRC stages I-III were approximately equally distributed (31.2%, 32%, and 26.4%, respectively), only 6.1% of participants had stage IV disease. Almost all participants had surgery (99.2%), 46.1% had chemotherapy, and 20.9% had radiotherapy. The start of rehabilitation was most frequently initiated up to 1 month after CRC surgery (42.3%) and only a few participants have had their surgery more than 1 year ago (3.9%). The latter are CRC patients who needed rehabilitation after a recurrence of the cancer and its treatment.

The majority of study participants had overweight or obesity (59.1%; BMI ≥ 25 kg/m^2^), were not engaged in a healthy level of physical activity prior to the CRC diagnosis (52.5%), and had two or more comorbidities (53.2%). Hypertension (52.7%) and diabetes mellitus (17.3%) were common, whereas few participants had a history of myocardial infarction or stroke (<4% each). Furthermore, 14.2% of study participants were current smokers. 

Almost half of the participants were fully employed (48.7%) prior to CRC treatment. More than one-third were in retirement (38.9%), while the rest were either part-time employed (9.3%) or unemployed (3.1%). 

### 3.2. Symptom Clusters

Symptom clusters were identified using exploratory factor analysis. The total number of items included in the analysis was 32. The Kaiser–Meyer–Olkin measure of sampling adequacy was 0.87. The Minimum Average Partial test suggested three factors as the optimal solution, while the Parallel Analysis suggested six factors. Using the criteria above, solutions with three, four, five, and six factors were explored. The six-factor solution was chosen because it explained the biggest proportion of the variance and had a clear conceptual structure. Apart from dry mouth, all symptoms had at least one factor loading greater than 0.30.

Factor 1 included fatigue, interference with daily life, emotional fatigue, depression, cognitive fatigue, dyspnea, social functioning, ability to work, and appetite loss. Factor 2 included fecal incontinence, stool frequency, sore skin, embarrassment, diarrhea, flatulence, and blood/mucus in stool. Factor 3 included pain, abdominal pain, buttock pain, bloating, and dysuria. Factor 4 included psychosocial stress, anxiety, and financial difficulties. Factor 5 included urinary frequency, urinary incontinence, and sleep disturbance. Factor 6 included taste alteration, hair loss, nausea or vomiting, and constipation. Upon examining the items loaded on each factor, we named factors 1 to 6 as fatigue factor, gastrointestinal symptoms factor, pain factor, psychosocial symptoms factor, urinary symptoms factor, and chemotherapy side effects factor, respectively. Factor loadings, eigenvalues, and explained variance can be found in Table 2. Since the oblique rotation was used, factors are correlated with each other (Appendix A). However, the correlations between factors are low-to-moderate, with the highest correlation between the fatigue factor and urinary symptoms factor (r = 0.56).

### 3.3. Longitudinal Analyses

#### 3.3.1. Longitudinal Association of Baseline Symptoms with Fatigue 

150 (54%) study participants had clinical fatigue (FACIT-F-FS < 34 points) at baseline and n = 82 (29.3%) at 3-month follow-up. The median FACIT-F-FS increased from 31 points (interquartile range (IQR): 22–42) to 40 points (IQR: 29–47) in this time, which translates to less fatigue. On average, the mean FACIT-F-FS improved by 5.9 points (standard deviation (SD): 9.7 points). The distribution of this change in FACIT-F-FS points was approximately normally distributed (Figure 1). 

The results of the multivariable linear regression analysis evaluating the association between covariates and the change in fatigue from baseline to 3-months follow-up are displayed in Appendix A. Having received chemotherapy or radiotherapy within one year before rehabilitation was the factor with the highest reduction in fatigue with a β coefficient (95% CI) of 4.71 (1.93; 7.48).

The association between baseline symptoms and the change in fatigue until the 3-month follow-up is displayed in Figure 2. In total, 14 out of 30 symptoms were significantly associated with the change in the fatigue score until the 3-month follow-up. Their ß coefficients were negative, indicating that subjects with these symptoms improved less in the extent of their fatigue symptoms than those without these symptoms. At least one of the symptoms from each factor was longitudinally associated with fatigue. Abdominal pain from the pain factor had the strongest association with fatigue, with a β coefficient (95% CI) of −5.11 (−7.86; −2.37). The β coefficients with 95% CIs and p values of all symptoms shown in Figure 2 can be found in Appendix A.

Subgroup analyses by age and sex showed several potential differences, which need to be interpreted with caution because all confidence intervals overlapped. We would like to highlight potential subgroup differences, with effect estimates that were statistically significant and among which one was at least twice as strong as another when comparing the subgroups. With respect to potential sex differences, sore skin, fecal incontinence, and all pain-related symptoms except abdominal pain, financial difficulties, psychological stress and hair loss were more strongly associated with the change in fatigue scores in females than in males (Appendix A). Dyspnea and social sequelae were more strongly associated with the change in fatigue scores in males than females. The subgroup analysis by age groups found that many symptoms from the gastrointestinal, pain and psychosocial symptom clusters were more strongly associated with the change in fatigue scores in study participants younger than 65 years than in older patients (Appendix A). Moreover, urinary incontinence and constipation were more strongly associated with the change in fatigue in younger study participants. In participants 65 years or older, only appetite loss was more strongly associated with the change in fatigue than among younger CRC patients (Appendix A). Subgroup analysis by participation in the VICTORIA trial did not show relevant differences in the results because all confidence intervals overlapped.). 

#### 3.3.2. Longitudinal Association of Baseline Symptoms with Ability to Work 

111 (40%) study participants had a low-to-moderate ability to work (FACIT-F-FWB-AW score ≤ 2 points) at baseline and 72 (25.8%) participants at 3-month follow-up. The median FACIT-F-FWB-AW increased from 3 points (interquartile range (IQR: 2–4) to 4 points (IQR: 2–4) in this time, which translates to higher ability to work. On average, the mean FACIT-F-FWB-AW score improved by 0.57 points (standard deviation (SD): 1.2 points). The distribution of this change in FACIT-F-FWB-AW score points was approximately normally distributed (Figure 3).

The results of the multivariable linear regression analysis evaluating the association between covariates and the change in the ability to work from baseline to 3-month follow-up are displayed in Appendix A. A higher baseline ability to work and having received chemotherapy or radiotherapy within the year before rehabilitation were significantly associated with improved ability to work at the 3-month follow-up, whereas all other factors were not associated with this outcome.

The association between baseline symptoms and the change in ability to work until the 3-month follow-up are displayed in Figure 4. Overall, 24 out of 30 symptoms were significantly associated with the change in ability to work until the 3-month follow-up. At least one of the symptoms from each factor was associated with the change in ability to work at the 3-month follow-up. Their ß coefficients were negative, indicating that subjects with these symptoms improved less in the ability to work than those without these symptoms. All of the symptoms stemming from the fatigue, pain, and psychosocial symptoms factors were associated with the ability to work at the 3-month follow-up. Nausea or vomiting had the strongest association with work ability, with a β coefficient (95% CI) of −0.83 (−1.12; −0.53). The β coefficients with 95% CIs and p values for all symptoms shown in Figure 4 can be found in Appendix A.

In the subgroup analyses on the ability to work, we focused again on potential subgroup differences, with statistically significant effect estimates, among which one was at least twice as strong compared to the other. However, it needs to be noted that all confidence intervals overlapped. The subgroup analysis by sex found that sore skin and dysuria were more strongly associated with the change in the ability to work among female than male participants. In contrast, several symptoms from the fatigue cluster (fatigue, appetite loss, social sequelae, and dyspnea) and diarrhea were more strongly associated with the change in the ability to work in male than in female CRC patients (Appendix A). Subgroup analysis by age showed comparable findings for the associations of most symptoms with the change in the ability to work with the exceptions of social sequelae, bloating, and constipation being more strongly associated with the change in the ability to work among patients younger than 65 years (Appendix A). Only dyspnea had a stronger effect estimate among older subjects. Subgroup analysis by participation in the VICTORIA study did not reveal differences in the results for the outcome “change in the ability to work”. 

## 4. Discussion

The primary objective of this study was to identify symptom clusters in CRC patients within a year of completing primary CRC treatment using exploratory factor analysis. The second aim was to investigate which symptoms were longitudinally associated with the change in fatigue or ability to work from baseline to the 3-month follow-up using multivariable linear regression. We identified six symptom clusters: fatigue, gastrointestinal symptoms, pain, psychosocial symptoms, urinary symptoms, and chemotherapy side effects clusters. Out of 32 symptoms with significant factor loadings, 14 symptoms were longitudinally associated with fatigue and 24 symptoms with the ability to work.

The fatigue cluster consisted of the following symptoms: fatigue (FACIT-F-FS), interference with daily life, emotional fatigue, depression, cognitive fatigue, dyspnea, social sequelae, appetite loss and ability to work. Fatigue is one of the most frequent symptoms of CRC patients, caused by primary disease or affiliated treatment. It is characterized as “*physical, social and/or cognitive tiredness or exhaustion related to cancer or cancer treatment that is not proportional to recent activity and interferes with functioning*” [31], and all of the symptoms included in this cluster, apart from dyspnea, could be interpreted as manifestations of cancer-related fatigue. Lack of appetite has been identified as a correlate of cancer-related fatigue [32]. Dyspnea, or the feeling of difficulty breathing, could be present during everyday physical activities such as walking or climbing the stairs. Cancer-related anemia, often present in CRC patients due to the chronic character of the disease or occult bleeding, can be a cause of both dyspnea and fatigue [33]. The fatigue cluster was often identified in previous studies on symptom clusters, irrespective of the type of malignancy [13,34]. In a network analysis of symptom clusters among a diverse group of cancer survivors, fatigue emerged as the most central symptom across all networks [35]. This finding highlights the crucial need to address fatigue early in the rehabilitation process.

The gastrointestinal symptoms cluster consisted of fecal incontinence, stool frequency, sore skin, embarrassment, diarrhea, flatulence, and blood/mucus in stool. Most of these symptoms can be directly linked to treatment side effects. Local radiotherapy can cause diarrhea, bloating, skin erythema, and fecal incontinence [10]. Due to the nature of these symptoms, patients can have accompanying embarrassment in social settings. The gastrointestinal symptoms cluster, often characterized as “digestive symptoms cluster”, is frequently identified in the previous literature in CRC patients [13,36] and patients with other cancer types [35]. 

The pain cluster consisted of pain, abdominal pain, buttock pain, bloating, and dysuria. Pain is one of the symptoms present at the time of the diagnosis and before treatment and can often be aggravated by subsequent treatments [37]. Oxaliplatin-induced neuropathy has emerged as a common side effect of chemotherapy for CRC patients [38]. Chronic pain can be a long-term side effect of CRC surgery. A Danish study found that 40% of CRC patients reported having chronic abdominal or pelvic pain after rectal surgery [39]. The pain cluster is commonly identified in cancer populations, irrespective of tumor localization [13,15,40].

The psychosocial symptoms cluster consisted of anxiety, financial difficulties, and psychosocial distress. During therapy, many patients need to take an absence from work. A previous study in the Netherlands found that CRC patients had a median return-to-work time of 423 days from the beginning of their absence leave [41]. The temporary absence from work can be a cause of financial difficulties as well as worry and anxiety for CRC patients. Anxiety is also frequent at the start of the treatment and can persevere after the completion of the treatment [42]. In some previous studies, anxiety has clustered with depression and other emotional symptoms [36,43]. However, in our study, anxiety coupled with psychosocial distress and financial difficulties. It is possible that the absence from work and its financial implications may have amplified the interconnectedness of anxiety, financial strain, and broader psychosocial distress in CRC patients in our study.

The urinary symptoms cluster consisted of urinary incontinence, urinary frequency, and sleep disturbance. Sleep disturbances are not unusual in patients with frequent urination during the night. A previous study found that late urinary side effects affect up to 41% of CRC patients following brachytherapy for colorectal cancer. Symptoms included obstructive and irritative symptoms [44]. The urinary symptoms have been described in previous studies of CRC symptom clusters. 

The chemotherapy side effects cluster included nausea or vomiting, taste alteration, constipation, and hair loss. These symptoms have been described as common post-chemotherapy side effects in the literature [45]. Though nausea and vomiting are sometimes grouped as a gastrointestinal cluster in the previous literature [46], they constitute their own cluster alongside other prevalent chemotherapy side effects in our study. These findings indicate the complexity of CRC symptomatology after treatment and variability in patient experience.

In the context of our findings, we identified six specific symptom clusters in CRC patients. Our findings point out that these clusters originate from the primary disease or, more predominantly, from therapeutic interventions. Notably, the fatigue and pain clusters, which are prominently present, suggest an underlying inflammatory mechanism, a facet that requires further exploration [47]. A recent study further supports this hypothesis by reporting an association between the symptom clusters in palliative CRC patients and proinflammatory cytokines [48]. This underscores the necessity to investigate potential inflammatory mechanisms underlying these symptom clusters, emphasizing the importance of ongoing research for improved management and comprehension of CRC-associated symptoms.

In this study, we identified two groups of symptoms to be longitudinally associated with fatigue: gastrointestinal/physical symptoms (fecal incontinence, stool frequency, urinary incontinence, abdominal pain, dyspnea, and nausea or vomiting) and psychosocial symptoms (cognitive fatigue, embarrassment, anxiety, financial difficulties, and sleep disturbance). Participants who experienced any of these symptoms were more likely to have higher fatigue levels. Gastrointestinal symptoms in female CRC patients were previously shown to be associated with fatigue, as well as reduced social functions and increased daily life interference [49]. Considering psychosocial factors, a cross-sectional study identified an association between fatigue and both sleep disturbances and anxiety in CRC patients [50]. However, while this study noted a significant relationship between fatigue and depression, our findings did not corroborate this link. Possible reasons for this inconsistency might include variations in study methodologies, patient demographics, or therapeutic regimens between the two studies. Recognizing symptoms associated with fatigue is crucial for clinical practice, allowing for more targeted interventions in fatigue management for CRC patients.

We found 24 symptoms that are longitudinally associated with the ability to work. Our findings confirm the results from the systematic review on the influence of cancer-related symptoms on the ability to work [19]. Previously, a study found that CRC patients had a 56% higher risk of losing employment due to work disability up to 4 years after CRC diagnosis, compared to the general population [51]. Symptoms related to lower ability to work stemmed from all six of the identified clusters. Out of 11 symptoms associated with improved fatigue at the 3-month follow-up, 9 were also associated with improved ability to work in the first 3 months after rehabilitation. These results reaffirm the pleiotropic influence of different symptoms on the ability to work in CRC patients, apart from the conventional predictors such as age and stage of the disease. It also underscores the importance of a holistic approach, emphasizing the need to address the entirety of symptoms rather than the traditional one-symptom focus. Understanding the clustering of symptoms and its connection to the ability to work is a central step in developing intervention strategies that could be implemented during the rehabilitation part of the convalescence of CRC patients. 

The MIRANDA study possesses several strengths. Firstly, it is a multicentric, prospective cohort study that aims to capture data from six rehabilitation clinics across different regions in Germany, ensuring a diverse and representative sample. The rigorous inclusion and exclusion criteria further ensure the consistency and reliability of the data. The study involves periodic follow-ups, which allows for a longitudinal understanding of the variables in question, such as fatigue and the ability to work. Furthermore, the study utilizes multiple well-established scales and questionnaires like the FACIT-F-FS, GDS-15, GAD-7, and EORTC-QLQ-C30, among others, to comprehensively assess various symptoms and factors.

However, there are some inherent limitations to our study. One limitation is the reliance on self-reported data for most sociodemographic, lifestyle, and clinical information, such as symptom presence. Only a few variables, such as weight, height, and stage of the disease, relied on physician reports. Additionally, the ability to work was estimated using a single-item measure. Various factors can influence the ability to work, including physical symptoms, cognitive function, environmental conditions, and social and emotional well-being. A single item might not capture all these dimensions. Lastly, almost half of the participants were recruited during the COVID-19 pandemic between Sept 2020 and Dec 2021. Consequently, there is a possibility that some of these participants might have experienced post-COVID symptoms or late effects from the virus infection [52]. This could potentially increase the prevalence of certain symptoms in our sample, making it higher than it might have been in a non-pandemic context. 

## 5. Conclusions

Our analysis identified six unique symptom clusters in CRC patients following curative treatment: fatigue, gastrointestinal symptoms, pain, psychosocial stress, urinary symptoms, and chemotherapy side effects. These clusters offer valuable insights for clinicians, suggesting a structured approach wherein symptoms can be addressed based on the cluster from which they originate. Notably, symptoms stemming from each of these clusters have shown an association with worse fatigue and lower ability to work at the 3-month follow-up.

Despite the efforts of current rehabilitation protocols, our findings indicate that several symptoms remain significantly associated with post-rehabilitation fatigue and a low ability to work in CRC patients. This highlights an area in our care continuum that requires further investigation and optimization. Enhanced interventions, integrating both pharmacological and psychosocial modalities, may provide improved outcomes for these patients. A targeted approach is recommended to address specific symptom clusters and their relationship with fatigue. Future studies should focus on developing and testing these interventions to optimize the post-treatment phase for CRC patients.

## Figures and Tables

**Figure 1 cancers-16-00202-f001:**
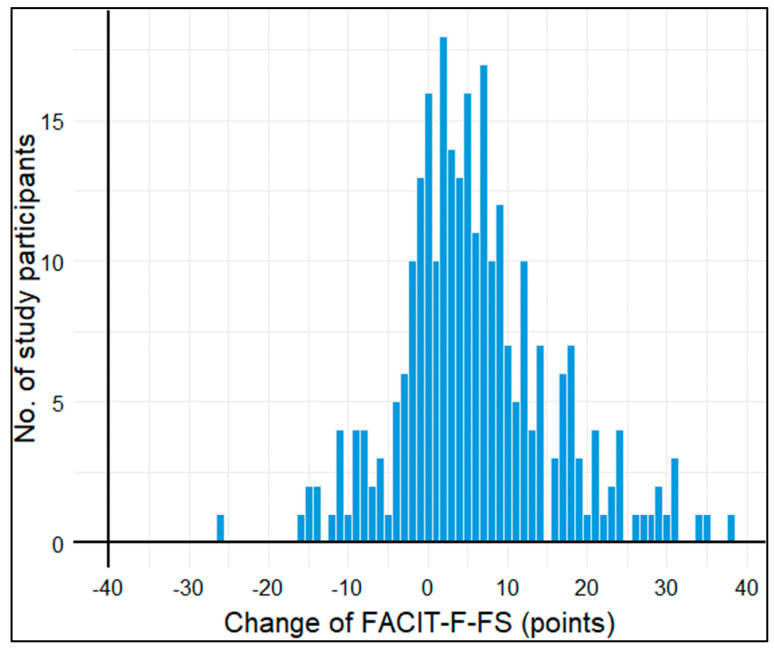
Histogram of change in FACIT−F−FS points from baseline to 3−month follow-up.

**Figure 2 cancers-16-00202-f002:**
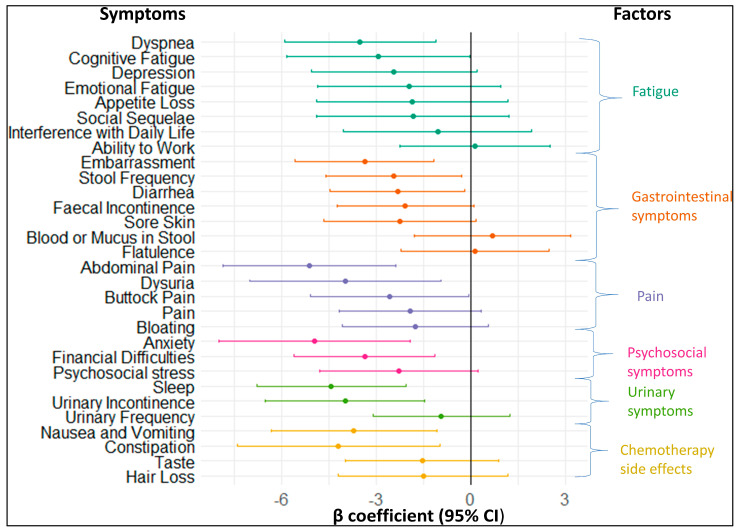
Forest plot showing longitudinal associations of symptoms at baseline with the change in fatigue symptoms from baseline to 3−month follow-up.

**Figure 3 cancers-16-00202-f003:**
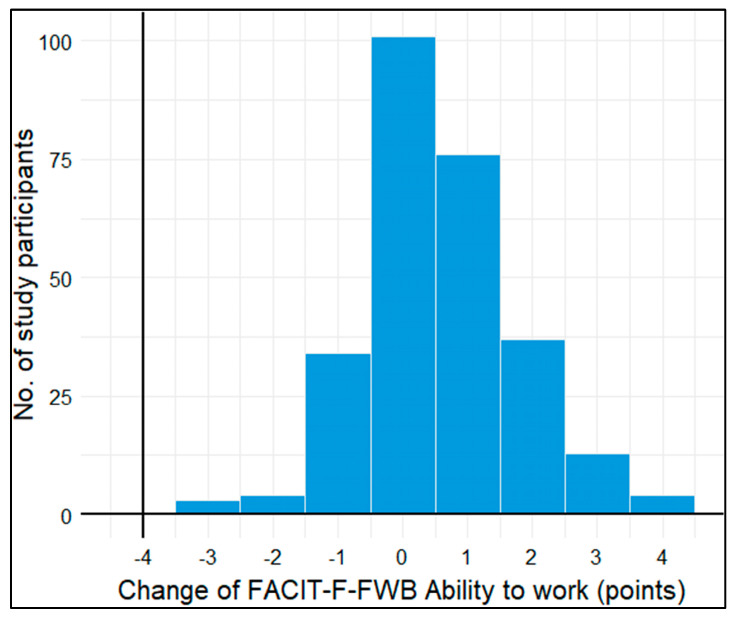
Histogram of change in FACIT-F-FWB ability to work score points from baseline to 3-month follow−up.

**Figure 4 cancers-16-00202-f004:**
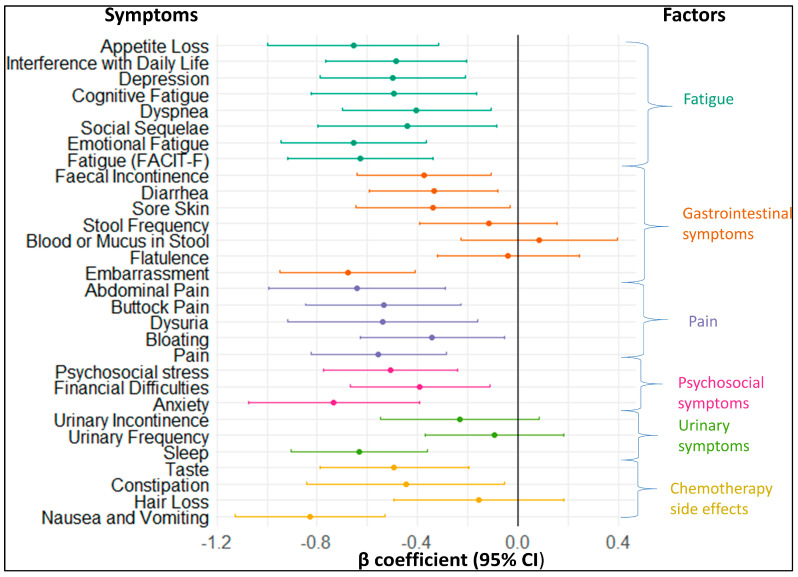
Forest plot showing longitudinal associations of symptoms at baseline with the change in the ability to work from baseline to 3−month follow-up.

**Table 1 cancers-16-00202-t001:** Baseline characteristics of the participants (N = 394).

Baseline Characteristics	N_total_	Proportion (%)	Median (Q1–Q3)
Age (years)	394		62 (56–71)
<65		222 (56.3)	
≥65		172 (43.7)	
Sex	394		
Female		166 (42.1)	
Male		228 (57.9)	
Cancer stage	375		
I		117 (31.2)	
II		120 (32.0)	
III		99 (26.4)	
IV		23 (6.1)	
Unknown		16 (4.3)	
Type of CRC treatment			
Surgery	389	386 (99.2)	
Chemotherapy	382	176 (46.1)	
Radiotherapy	378	79 (20.9)	
Months since CRC surgery	383		
0–1		162 (42.3)	
2–3		62 (16.2)	
4–6		48 (12.5)	
7–9		62 (16.2)	
10–12		34 (8.9)	
>12		15 (3.9)	
Body mass index (kg/m^2^)	394		26.2 (23.1–29.6)
<25		161 (40.9)	
25 to <30		144 (36.5)	
≥30		89 (22.6)	
Smoking status	373		
Never smoked		159 (42.6)	
Former smoker		161 (43.2)	
Current smoker		53 (14.2)	
Healthy physical activity level ^a^	368	175 (47.5)	
Comorbidities			
Diabetes mellitus	371	64 (17.3)	
Hypertension	368	194 (52.7)	
History of myocardial infarction	370	13 (3.5)	
History of stroke	371	14 (3.8)	
Number of comorbidities	374		2 (1–2)
0		66 (17.7)	
1		109 (29.1)	
≥2		199 (53.2)	
Employment status	386		
Fully employed		188 (48.7)	
Part-time employed		36 (9.3)	
Retired		150 (38.9)	
Unemployed		12 (3.1)	

Abbreviations: CRC, colorectal cancer; Q1, 1st quartile (25th percentile); Q3, 3rd quartile (75th percentile). ^a^ healthy physical activity was characterized as engaging in a minimum of 150 minutes of moderate-intensity or 75 min of high-intensity aerobic exercise weekly, or a comparable mix of moderate activities, in the year leading up to the diagnosis of CRC.

**Table 2 cancers-16-00202-t002:** Allocation of 32 colorectal cancer symptoms to six factors.

Symptoms	Factor 1 “Fatigue”	Factor 2 “Gastro- Intestinal Symptoms”	Factor 3 “Pain”	Factor 4 “Psycho- Social Symptoms”	Factor 5 “Urinary Symptoms”	Factor 6 “Chemotherapy Side Effects”
	Highest Factor Loading					
Fatigue (FACIT F)	−0.97					
Interference with Daily Life	0.94					
Emotional Fatigue	0.80					
Depression	0.68					
Cognitive Fatigue	0.61					
Dyspnea	0.49					
Social Functioning	0.46					
Ability to Work	−0.39					
Appetite Loss	0.31					
Fecal Incontinence		0.82				
Stool Frequency		0.80				
Sore Skin		0.65				
Embarrassment		0.51				
Diarrhea		0.45				
Flatulence		0.36				
Blood/Mucus in Stool		0.33				
Abdominal Pain			0.79			
Pain			0.73			
Buttock Pain			0.52			
Bloating			0.43			
Dysuria			0.35			
Psychosocial Stress				0.58		
Anxiety				0.48		
Financial Difficulties				0.40		
Urinary Frequency					0.70	
Urinary Incontinence					0.50	
Sleep Disturbance					0.33	
Dry Mouth					0.20	
Taste Alteration						0.61
Hair Loss						0.44
Constipation						0.32
Nausea or Vomiting						0.30
Eigenvalue	4.96	2.67	2.17	1.99	1.12	0.95
Variance explained	0.15	0.08	0.07	0.06	0.04	0.03
Proportion explained (%)	36	19	16	14	8	7

## Data Availability

The data will not be published on an open-access platform. After completion of the study, interested scientists can request data use and receive pseudonymized data upon the approval of this application by the principal investigator. Please contact Ben Schöttker (b.schoettker@dkfz.de).

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
