# Peer review of "From a Clustering of Adverse Symptoms after Colorectal Cancer Therapy to Chronic Fatigue and Low Ability to Work: A Cohort Study Analysis with 3 Months of Follow-Up"

_cancers, 2024, doi:10.3390/cancers16010202_

Round 1
Reviewer 1 Report
Comments and Suggestions for Authors
please review the attached documents as there are some clarifications required
this work is very interesting and certainly an area to investigate further

Comments on the Quality of English Languageonly minor details see attached document
Author Response
We thank the reviewer for the helpful comments. We have modified our manuscript accordingly. Attached you will find our point-by-point response.

Reviewer 2 Report
Comments and Suggestions for Authors
Dear authors,
thank you for the opportunity to review your work.
The research is very interesting, up to date, novel, and extremely important, especially for planning psychosocial interventions for patients with CRC.
Your manuscript is very well organized, in the introductory part all relevant and recent findings are explained.
The sample selection is appropriate and representative.
The methodology is well conceived and allows for the implementation of statistical procedures. Relevant statistical procedures and data processing were used.
In the discussion part, you gave a clear presentation of the obtained results and compared them with similar recent research.
Well done!
Author Response
We thank the reviewer for the appraisal of our manuscript.
